# Can Graph Neural Networks Expose Training Data Properties? An Efficient Risk Assessment Approach

**Hanyang Yuan**[1,2,4†‡], **Jiarong Xu**[2*], **Renhong Huang**[1]
**Mingli Song**[1,4‡], **Chunping Wang**[3], **Yang Yang**[1]
[1]Zhejiang University, [2]Fudan University, [3]Finvolution Group
[4]Hangzhou High-Tech Zone (Binjiang) Institute of Blockchain and Data Security
{yuanhanyang, renh2, brooksong, yangya}@zju.edu.cn
jiarongxu@fudan.edu.cn, wangchunping02@xinye.com

## Abstract

Graph neural networks (GNNs) have attracted considerable attention due to their diverse applications. However, the scarcity and quality limitations of graph data present challenges to their training process in practical settings. To facilitate the development of effective GNNs, companies and researchers often seek external collaboration. Yet, directly sharing data raises privacy concerns, motivating data owners to train GNNs on their private graphs and share the trained models. Unfortunately, these models may still inadvertently disclose sensitive properties of their training graphs (*e.g.*, average default rate in a transaction network), leading to severe consequences for data owners. In this work, we study graph property inference attack to identify the risk of sensitive property information leakage from shared models. Existing approaches typically train numerous shadow models for developing such attack, which is computationally intensive and impractical. To address this issue, we propose an efficient graph property inference attack by leveraging model approximation techniques. Our method only requires training a small set of models on graphs, while generating a sufficient number of approximated shadow models for attacks. To enhance diversity while reducing errors in the approximated models, we apply edit distance to quantify the diversity within a group of approximated models and introduce a theoretically guaranteed criterion to evaluate each model's error. Subsequently, we propose a novel selection mechanism to ensure that the retained approximated models achieve high diversity and low error. Extensive experiments across six real-world scenarios demonstrate our method's substantial improvement, with average increases of 2.7% in attack accuracy and 4.1% in ROC-AUC, while being 6.5× faster compared to the best baseline.

## 1 Introduction

Graph data, encapsulating relationships between entities across various domains such as social networks, molecular networks, and transaction networks, holds immense value [1–3]. Graph neural networks (GNNs) have proven effective in modeling graph data [4–6], yielding promising results across diverse applications, including recommender systems [7], molecular prediction [8, 9], and anomaly detection [10]. While training high-quality GNN models may necessitate a substantial amount of data, graphs may be scarce or of low quality in practice [11, 12], prompting companies and researchers to seek additional data from external sources [13].

---

[†]This work was done when the author was a visiting student at Fudan University.
[‡]State Key Laboratory of Blockchain and Security, Zhejiang University.
[*]Corresponding author.

38th Conference on Neural Information Processing Systems (NeurIPS 2024).

However, directly obtaining data from other sources is often difficult due to privacy concerns [14, 15]. As an alternative, sharing models rather than raw data has become increasingly common [13]. Typically, *data owners* train a model on their own data and subsequently release it to the community or collaborators [16–20]. For instance, a larger bank may train a fraud detection model on its extensive transaction network and share it with partners, allowing them to use their own customer data to identify risks.

Despite the benefits, this model-sharing strategy sometimes remains vulnerable to data leakage risks. Given access to the released model, one may infer sensitive properties of the data owner's graph, which are not intended to be shared. In the context of releasing a fraud detection model, if an adversarial bank can determine the average default rate of all customers in the transaction network, the data owner bank's financial status can potentially be revealed. Another example is releasing a recommendation model trained on a company's product network [21]. If a competitor can infer the distribution of co-purchase links between different products, he may determine which items are frequently promoted together and deduce the company's marketing tactics. Such attacks are possible because released models may inadvertently retain and expose sensitive information from the training data [22, 23]. We refer to such sensitive information related to the global distribution in a graph as *graph sensitive properties*, and we aim to investigate the problem of *graph property inference attack*.

Previous property inference attacks [22–25] primarily focus on text or image data, assuming models trained on different properties exhibit differences in parameters or outputs. For GNNs modeling graph data, the inherent relationships and message-passing mechanisms can magnify distribution bias [26], making them more vulnerable to attacks. Although a few studies extend property inference to graphs and GNNs [20, 21, 27], they typically involve creating shadow models that replicate the released model's architecture and are trained on shadow graphs with varying sensitive properties. The parameters or outputs of shadow models are used to train an attack model to classify the property of the data owner's graph. A major limitation of these attacks is the need to train a large number of shadow models (*e.g.*, 4,096 models [22], 1,600 models [27]), resulting in significant computational cost and low efficiency.

In this paper, we explore the feasibility of avoiding the training of numerous shadow models by designing an *efficient yet effective* graph property inference attack. Our key insight is to train only a small set of models and then generate sufficient approximated shadow models to support the attack. To this end, we leverage and extend model approximation techniques. For a given dataset and a model trained on it, when the training data changes (*e.g.*, removing a sample), model approximation allows the efficient estimation of new model parameters for the updated dataset without retraining. This technique, often called unlearning [28–31], enables the efficient generation of multiple approximated shadow models from a single trained model. Specifically, given a small set of graphs and their corresponding trained models, we perturb each graph to alter sensitive properties (*e.g.*, changing the number of nodes corresponding to high default rate users) and then apply model approximation to produce a sufficient number of approximated models corresponding to the perturbations, thereby reducing the total attack cost. Figure 1 illustrates our approach compared to the traditional attack.

Nevertheless, achieving this goal presents several challenges. The first challenge is to ensure the diversity of approximated models, which provides a broader range of training samples for the attack model and enhances its generalization capability. To tackle this, we develop structure-aware random walk sampling graphs from distinctive communities and introduce edit distance to quantify the diversity of a set of approximated models. The second challenge is to ensure that the errors in the approximated shadow models are sufficiently small. Otherwise, these models may fail to accurately reflect differences in graph properties, thereby diminishing attack performance. To address this, we establish that different graph perturbations can lead to varying approximation errors, which offers a theoretical-guaranteed criterion for assessing the errors of each approximated model. Finally, we propose a novel selection mechanism to reduce errors while enhancing the diversity of approximated models, formulated as an efficiently solvable programming problem. Our contributions are as follows:

- We propose an efficient and effective graph property inference attack that requires training only a few models to generate sufficient approximated models for the attack.

- We propose a novel selection mechanism to retain approximated models with high diversity and low error, using edit distance to measure the diversity of approximated models and a theoretical criterion for assessing the errors of each. This diversity-error optimization is formulated as an efficiently solvable programming problem.

- Experiments on six real-world scenarios demonstrate the efficiency and effectiveness of our proposed attack method. On average, existing attacks require training 700 shadow models to achieve 67.3% accuracy and 63.6% ROC-AUC, whereas our method trains only 66.7 models and obtains others by approximation, achieving 69.0% attack accuracy and 66.4% ROC-AUC.

## 2  Problem Definition

The scenario of a graph property inference attack first involves a data owner who trains a GNN model using his graph data, referred to as the *target model* and *target graph* in the following text. Once trained, the target model's parameters or output posterior probabilities can be released in communities [16, 18]. For example, the data owner can upload the pre-trained parameters to GitHub [17, 19] to facilitate downstream tasks. Or he may upload the posterior probabilities from a recommender system to a third-party online optimization solver [21], such as Gurobi *. Collaborative machine learning is another potential attack scenario [20]. For instance, consider two banks aiming to collaboratively train a fraud detection model. While sharing raw transaction networks poses risks to privacy and commercial confidentiality, they can employ model-sharing strategies [13].

With access to the target model, curious users may launch inference attacks to obtain some sensitive properties of the target graph, which can reveal secrets not intended to be shared. For example, the ratio of co-purchase links between particular products in a product network may relate to the promoting tactics of a sales company, or the average default rate in a transaction network may reveal the financial status of a bank. Such confidential information may further impact business competition.

In the rest of this section, we first define the privacy, *i.e.*, the sensitive properties referred to in this work. Then, we introduce the knowledge of the attack. Finally, we formulate the problem of property inference attacks.

**Graph sensitive property.**   In this paper, we consider an attributed target graph where nodes are associated with multiple attributes. The sensitive property is defined based on one specific type of attribute, called the *property attribute*. Specifically, the sensitive property is defined as a certain statistical value of the property attribute's distribution. We consider two types of properties that the attacker may infer: (1) node properties, specified by the ratio of nodes with a particular property attribute value, and (2) link properties, specified by the ratio of links where the end nodes have particular property attribute values.

Note that the property attribute can be either discrete or continuous. For instance, a node property defined on the discrete category attribute in a product network can be the ratio of co-purchase links between luxury items. An edge property defined on the continuous default rate attribute in a transaction network can be the average default rate of all customers. The inferred sensitive properties may reveal data owner's secrets such as commercial strategies; see [23, 32] for further discussion.

**Attacker's knowledge.**   The attacker's background knowledge is assumed to be as follows:

- Auxiliary graph: We assume the attacker has an auxiliary graph from the same domain as the target graph but does not necessarily intersect with the latter. In practice, the auxiliary graph can be sourced from publicly available data or derived directly from the adversary's own knowledge.
- Target model: We consider two types of knowledge on the target model: the white-box setting, where the adversary knows the architecture and parameters of the target GNN, and the black-box setting, where the adversary only knows the target GNN's output posterior probabilities.

**Property inference attack.**   Formally, let $G^{\text{tar}}$ denote the target graph. And let $\mathcal{P}(G^{\text{tar}})$ denote the property value of $G^{\text{tar}}$. Note that $\mathcal{P}$ can represent either node properties or link properties. Given that the attacker has an auxiliary graph $G^{\text{aux}}$ from the same domain as $G^{\text{tar}}$, we define graph property inference attack as follows:

**Problem 1 (Graph property inference attack)** *Given the auxiliary graph $G^{\text{aux}}$, and assume the attacker has either the white-box knowledge of the target GNN parameters or the black-box knowledge of target GNN's output posterior probabilities, the objective of the graph property inference attack is to infer the property $\mathcal{P}(G^{\text{tar}})$ without access to it.*

---

*  https://www.gurobi.com

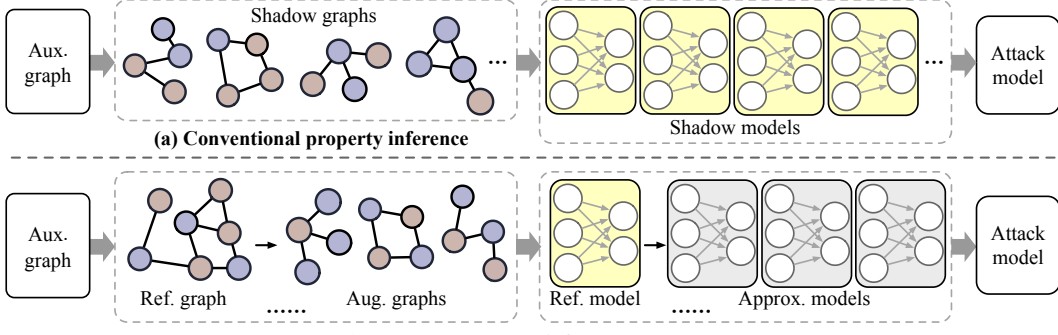

Figure 1: Illustrations of (a) conventional graph property inference attacks and (b) the proposed attack, with yellow shading indicating model training, the main source of computational cost.

# 3 Methodology

This section provides a detailed description of the proposed graph property inference attack. We start with an overview of our method and then delve into the technical aspects of model approximation. Finally, we describe how to ensure the diversity of approximated models while reducing their errors. The overall algorithm and complexity analysis are summarized in Appendix A.3.

## 3.1 Overview

As shown in Figure 1(a), given the auxiliary graph, the conventional approach is to first sample numerous shadow graphs, ensuring that graphs with different properties are adequately represented. Each is then used to train a shadow model with the same structure as the target model. Once trained, parameters (white-box) or output posterior probabilities (black-box) of shadow models are collected, along with the corresponding properties of shadow graphs. Finally, an attack model (*e.g.* linear classifier) is trained to classify properties based on parameters or posteriors. Since the number of shadow models is usually hundreds or thousands, their training can be computationally expensive.

To mitigate this issue, our proposed attack method utilizes model approximation techniques as a substitute. We provide a illustration of our method in Figure 1(b):

(1) Instead of numerous shadow graphs, we first sample only a few *reference graphs*.

(2) On each reference graph, we train a *reference model* with the same architecture as the target model, and generate multiple augmented graphs by removing different nodes and edges.

(3) By efficient model approximation, we obtain approximated models *w.r.t* to the augmented graphs.

(4) We collect parameters or posteriors of all approximated models and train the attack model in a similar manner as previous attacks.

Here, we mainly face two challenges: ensuring that the approximate error associated with augmentations is relatively small, and ensuring that the approximated models are sufficiently diverse. To address them, we derive a theoretical criterion for calculating approximation errors across different augmented graphs (see § 3.2) and design a diversity enhancement strategy in § 3.3.

## 3.2 Model approximation and error analysis

We proceed by introducing the techniques of model approximation, which include generating augmented graphs, obtaining approximated models, and conducting theoretical analysis for error criterion.

**Generating augmented graphs and identifying influenced nodes.** First, we aim to ensure that multiple perturbations produce distinctive augmented graphs. This is essential because highly similar augmentations reduce the distinction in the corresponding graph properties and model features, providing minimal benefit to the overall attack. For this purpose, we propose removing both nodes and edges from the reference graph. Formally, Let the reference graph be denoted by $G^{\text{ref}} = (V, E)$ sampled from $G^{\text{aux}}$, where $V$ is the node set, $E \subseteq V \times V$ is the edge set. For one perturbation, we

remove $V^{\mathrm{R}} \subset V$ and $E^{\mathrm{R}} \subset E$ to obtain the augmented graph $G^{\mathrm{aug}}$. In GNNs, the neighborhood aggregation makes the removal inevitably influence the state of other remaining nodes. Given a $l$-layer GNN, the influenced nodes of removing a single node $v \in V^R$ is the $l$-hop neighborhood of $v$, denote as $\mathcal{N}_l(v)$. And the influenced nodes of removing a single edge $e \in E^{\mathrm{R}}$, connecting nodes $v$ and $u$, is denoted as $\mathcal{N}_l(e) = \mathcal{N}_{l-1}(v) \cup \mathcal{N}_{l-1}(u) \cup \{v, u\}$. With these in mind, we next define the total influenced nodes for removing $V^{\mathrm{R}}$ and $E^{\mathrm{R}}$.

**Definition 1 (Influenced nodes )** *Given the removed nodes $V^{\mathrm{R}}$, removed edges $E^{\mathrm{R}}$ and a $l$-layer GNN, the total influenced nodes $V^{\mathrm{I}}$ is defined as*

$$V^{\mathrm{I}} = \bigcup_{e \in E^{\mathrm{R}}} \mathcal{N}_l(e) \bigcup_{v \in V^{\mathrm{R}}} \mathcal{N}_l(v). \tag{1}$$

*Note that $V^{\mathrm{I}}$ is exclusive of $V^{\mathrm{R}}$; we omit the set difference for simplicity.*

**Generating Approximated Models.** Subsequently, we generate the approximated model based on the perturbation. While existing graph unlearning [29–31, 33] may offer potential solutions, they are either limited to specific model architectures or only support the removal of nodes or edges individually, making them unsuitable for direct application. To address this, we extend their mechanisms to suit our scenario. Let the reference model be parameterized by $\theta^{\mathrm{ref}} \in \mathbb{R}^m$. In this paper, we consider cross-entropy loss as the loss function, and $\theta^{\mathrm{ref}}$ is obtained as follows:

$$\theta^{\mathrm{ref}} = \arg \min_{\theta} \sum_{v \in V} \ell(\theta; v, E). \tag{2}$$

After removing $V^{\mathrm{R}}$ and $E^{\mathrm{R}}$, directly retraining on $G^{\mathrm{aug}}$ could yield a new model parameter $\theta^{\mathrm{aug}}$:

$$\theta^{\mathrm{aug}} = \arg \min_{\theta} \sum_{v \in V/V^{\mathrm{R}}} \ell(\theta; v, E/E^{\mathrm{R}}). \tag{3}$$

To avoid training from scratch, we derive the approximation of $\theta^{\mathrm{aug}}$ by the following theorem.

**Theorem 3.1 (GNN model approximation)** *Given the GNN parameter $\theta^{\mathrm{ref}}$ on $G^{\mathrm{ref}}$, the removed nodes $V^{\mathrm{R}}$, removed edges $E^{\mathrm{R}}$ and influenced nodes $V^{\mathrm{I}}$. Assume $\ell$ is twice-differentiable everywhere and convex, we have*

$$\theta^{\mathrm{aug}} \approx \theta^{\mathrm{ref}} + (\nabla^2 \mathcal{L}(\theta^{\mathrm{ref}}; G^{\mathrm{aug}}))^{-1} \nabla \big( \sum_{v \in V^{\mathrm{I}} \cup V^{\mathrm{R}}} \ell(\theta^{\mathrm{ref}}; v, E) - \sum_{v \in V^{\mathrm{I}}} \ell(\theta^{\mathrm{ref}}; v, E/E^{\mathrm{R}}) \big), \tag{4}$$

*where $\nabla$ denote gradient, and $\nabla^2$ denote Hessian. $\mathcal{L}(\theta^{\mathrm{ref}}; G^{\mathrm{aug}}) = \sum_{v \in V/V^{\mathrm{R}}} \ell(\theta^{\mathrm{ref}}; v, E/E^{\mathrm{R}})$. The detailed derivation can be found in Appendix A.2.*

In practice, the Hessian may be non-invertible due to the non-convexity of GNNs. We address this by adding a damping term to the Hessian [34]. To reduce computation, we also follow [29] to convert the inverse Hessian calculation into quadratic minimization. See Appendix A.3 for complexity analysis.

**Analyzing the approximation error.** Eventually, we aim to quantitatively assess the error in the approximated model, as this directly determines whether graph properties can be effectively reflected, thereby influencing the attack. To achieve this, we investigate how specific removal choices of $V^{\mathrm{R}}$ and $E^{\mathrm{R}}$ affect the approximation error in Eq. (4). Note that $\nabla \sum_{v \in V/V^{\mathrm{R}}} \ell(\theta; v, E/E^{\mathrm{R}}) = 0$ only when $\theta^{\mathrm{aug}}$ is the exact minimizer, thus the gradient norm $\|\nabla \sum_{v \in V/V^{\mathrm{R}}} \ell(\theta; v, E/E^{\mathrm{R}})\|_2$ can reflect the approximation error. The following theorem provides an upper bound on this gradient norm.

**Theorem 3.2 (Approximation error bound)** *Assume $\ell$ is twice-differentiable everywhere and convex, $\|\nabla \ell\|_2 \le c_1$, $\nabla^2 \sum_{v \in V/V^{\mathrm{R}}} \ell(\theta; v, E/E^{\mathrm{R}})$ is $\gamma_1$-Lipschitz, the approximation error bound is given by:*

$$\|\nabla \sum_{v \in V/V^{\mathrm{R}}} \ell(\theta^{\mathrm{aug}}; v, E/E^{\mathrm{R}})\|_2 \le C(|V^{\mathrm{R}}| + 2|V^{\mathrm{I}}|)^2 = C \cdot \delta(V^{\mathrm{R}}, E^{\mathrm{R}}), \tag{5}$$

*where $|\cdot|$ denotes the cardinality of a set, and $\delta(\cdot, \cdot)$ denotes the square of the number of nodes removed and influenced, given $V^{\mathrm{R}}$ and $E^{\mathrm{R}}$. $C$ denotes a constant depending on the GNN model, see Appendix A.2 for detail proof.*

Theorem 3.2 indicates that the error bound for the approximation is related to both the number of removed nodes and influenced nodes. Next, we demonstrate how this can serve as an *error criterion* to select augmented graphs that result in minimal approximation errors.

## 3.3 Diversity enhancement

Following the above, we detail the designed diversity enhancement strategy. To develop a well-generalized attack model capable of distinguishing different sensitive properties, we first apply a structure-aware random walk for sampling diverse reference graphs. We then propose a novel selection mechanism to ensure that multiple perturbations on the reference graphs further enhance diversity while considering the reduction of approximation error.

**Sampling diverse reference graphs.** Inspired by community detection, where diverse communities are identified on a graph, we design a structure-aware random walk for sampling reference graphs. Specifically, we incorporate Louvain community detection [35] to partition the auxiliary graph into several similarly sized communities. During random walks, the starting nodes are chosen from different communities. And we assign different weights to neighboring nodes: $w$ for those within the same community and $1 - w$ for those from different communities, where $w \in [0, 1]$ is a hyper-parameter. The transition probabilities are then obtained by normalizing these weights. This strategy encourages sampling reference graphs within distinct communities, thus boosting their diversity.

**Ensuring diverse augmented graphs.** To ensure perturbations on reference graphs can enhance the diversity, we further design a perturbation selector. Based on § 3.2, it is easy to see that each approximated model can be considered as a result of the specific perturbation. Thus, improving the diversity of approximated models is essentially improving the diversity of augmented graphs. Formally, for each reference graph we generate $k$ augmented graphs $\mathcal{G}^{\mathrm{aug}} = \{G_1^{\mathrm{aug}}, G_2^{\mathrm{aug}}, \ldots, G_k^{\mathrm{aug}}\}$ by randomly removing $k$ different sets of nodes and edges. The diversity for $\mathcal{G}^{\mathrm{aug}}$ is defined as:

**Definition 2 (Diversity for $\mathcal{G}^{\mathrm{aug}}$)** *Given a set of $k$ graphs $\mathcal{G}^{\mathrm{aug}} = \{G_1^{\mathrm{aug}}, \ldots, G_k^{\mathrm{aug}}\}$, and a graph metric $d(G_i^{\mathrm{aug}}, G_j^{\mathrm{aug}})$ that measures the distance between $G_i^{\mathrm{aug}}$ and $G_j^{\mathrm{aug}}$. The diversity of $\mathcal{G}^{\mathrm{aug}}$ is defined as the sum of all pair-wise graph distances in $\mathcal{G}^{\mathrm{aug}}$, that is, $\sum_{i=1}^{k} \sum_{j=1}^{k} d\left(G_i^{\mathrm{aug}}, G_j^{\mathrm{aug}}\right)$.*

Since stochastic augmentations may not all contribute to total diversity, our objective is to select a diverse subset of $\mathcal{G}^{\mathrm{aug}}$, namely, a subset of diverse perturbations to enhance the diversity of augmented models. However, it is important to note that solely maximizing diversity may lead to relatively large approximation errors, which may worsen the attack performance. Fortunately, utilizing the error criterion from Eq. (5), we can ensure that augmentations enhance diversity while minimizing total approximation error, which can be formulated as a quadratic integer programming task.

Given $k$ available perturbations, we aim to select $q$ of them, such that the diversity among these selected is maximized while keeping the approximation error minimal. We here introduce decision variables $x_i \in \{0, 1\}$ to represent whether the $i$-th augmentation is selected. Let $\delta_i$ represent the approximation error in the $i$-th augmentation (*cf.* Eq. (5)). The optimization problem is as follows:

$$\min \sum_{i=1}^{k} \sum_{j=1}^{k} d\left(G_i^{\mathrm{aug}}, G_j^{\mathrm{aug}}\right) x_i x_j, \quad \text{s.t. } (1) \sum_{i=1}^{k} x_i = q, (2) \sum_{i=1}^{k} \delta_i x_i \leq \epsilon, \tag{6}$$

where $\epsilon$ is a constant that imposes the budget on the total approximation error of the selected $q$ augmentations, ensuring that it does not exceed $\epsilon$. Here, we select graph edit distance as the distance metric, which can be efficiently calculated since all $k$ augmented graphs $\mathcal{G}^{\mathrm{aug}}$ are derived from one reference $\mathcal{G}^{\mathrm{ref}}$. We utilize Gurobi Optimizer [36], a state-of-the-art solver, to solve this quadratic integer programming problem, which is known for its efficiency and effectiveness.

## 3.4 Overall algorithm

We summarize the overall algorithm of our attack in Algorithm 1. Steps 1-2 outline the structure-aware random walk for sampling reference graphs, steps 5-12 detail the perturbation selector, with step 10 calculating the error criterion in Eq. (5). Finally, in step 16 we train the attack model.

Table 1: Properties to be attacked, # indicates number of nodes or edges.

| Type | Dataset | Property attribute | Property description |
|------|---------|--------------------|-----------------------|
| Node | Pokec | Gender | # male users > # female users |
| | Facebook | Gender | # male users > # female users |
| | Pubmed | Keyword "IS" | # publications with "IS" > # publications w/o "IS" |
| Link | Pokec | Gender | # same-gender edges > # diff-gender edges |
| | Facebook | Gender | # same-gender edges > # diff-gender edges |
| | Pubmed | Keyword"IS" | # edges between papers with "IS" > # other edges |

## 4 Experiments

In this section, we evaluate the performance of the proposed attack method by addressing the following three research questions:

- **RQ1**: How efficient and effective is our method on various graph datasets?

- **RQ2**: How do different factors influence the performance of our method?

- **RQ3**: How applicable is our method in different scenarios?

### 4.1 Experimental setup

**Datasets and sensitive properties.** We conduct property inferences on three real world datasets: Facebook [37], Pubmed [38], and Pokec [39]. Appendix A.4 details the datasets and properties.

- Facebook and Pokec are social networks where nodes represent users and edges denote friendships. Following [21], we select gender as the property attribute, set node property as whether the male nodes are dominant, and edge property as whether the same-gender edges are dominant.

- Pubmed is a citation network where nodes are publications and edges are citations. We select the keyword "Insulin" (IS) as the property attribute. Node property is whether publications with "IS" are dominant. Edge property is whether citations between publications with "IS" are dominant. All used properties are summarized in Table 1.

**Training and testing data.** For fairness, we evaluate our method and baselines on the same target graphs. To ensure there is no overlap between the target graph and the auxiliary graph, for each dataset we first use Louvain community detection to split the original graph into two similarly sized parts. One part is used as the auxiliary graph, and the other part is used to sample multiple target graphs. Sizes and numbers of reference graphs (our method), shadow graphs (baselines), and target graphs are provided in Appendix A.4.

**Target GNN.** For target GNN, We use a widely recognized GNN model, GraphSAGE [40], configured as per [21] with 2 layers, 64 hidden sizes, and 1,500 training epochs with an early stop tolerance of 50. The Adam optimizer is used with a learning rate of 1e-4 and a weight decay of 5e-4.

**Implementation details.** For the attack model, We use a linear classifier with the deepest trick [41]; For hyper-parameter settings, we perform grid searches of reference graphs' numbers in (0, 100] (step size 25), and augmented graphs' numbers in (0, 10] (step size 2), across all datasets. Experiments are repeated 5 times to report the averages with standard deviations. See appendix A.4 for more details. Our codes are available at `https://github.com/zjunet/GPIA_NIPS`.

**Baselines.** We adopt four state-of-the-art baseline models to compare against the proposed attack model: (1) GPIA [21]: An attack method designed for graphs and GNNs in both white-box and black-box settings, following the traditional attack framework. (2) PIR-S/PIR-D [22]: Two permutation equivalence methods designed for white-box attacks, PIR-S using neuron sorting and PIR-D using set-based representation. (3) AIA [42]: Property inference method based on attribute inference attack, which first predicts the property attribute based on embeddings/posteriors and then predicts property, suitable for both white-box/black-box attacks. See Appendix A.4 for details.

Table 2: Average accuracy and runtime (seconds) comparison on different properties in white-box setting. "Node" and "Link" denote node and link properties, respectively. The best results are in bold.

| | Facebook | | | | Pubmed | | | | Pokec | | | |
| | Node | | Link | | Node | | Link | | Node | | Link | |
| | Acc. | Rt. | Acc. | Rt. | Acc. | Rt. | Acc. | Rt. | Acc. | Rt. | Acc. | Rt. |
|---|---|---|---|---|---|---|---|---|---|---|---|---|
| GPIA | 60.7 | 1791 | 56.1 | 1591 | 82.9 | 3028 | 73.8 | 2901 | 58.6 | 1176 | 54.7 | 1314 |
| PIR-D | 67.8 | 1733 | 58.1 | 1563 | 83.9 | 3022 | 73.9 | 2895 | 60.5 | 1187 | 56.2 | 1325 |
| PIR-S | 60.2 | 1762 | 57.7 | 1576 | 80.5 | 3032 | 72.6 | 2912 | 59.6 | 1181 | 57.9 | 1318 |
| AIA | 64.3 | 1741 | 56.7 | 1553 | 67.3 | 3026 | 75.5 | 2896 | 58.3 | 1215 | 56.5 | 1353 |
| Ours | **71.4** | **254** | **61.5** | **222** | **85.2** | **550** | **75.9** | **432** | **61.4** | **242** | **58.8** | **159** |

## 4.2 Evaluation of efficiency and effectiveness (RQ1)

We first focus on white-box settings and evaluate the accuracy and ROC-AUC for effectiveness and runtime for efficiency. Note that the reported runtime throughout this work encompasses the entire attack process for both the proposed method and baselines, starting from sampling the reference (shadow) graphs to inferring the properties of the target graphs. Table 2 presents the average accuracy and runtime of the proposed attack method compared to other baseline methods on the six aforementioned sensitive properties. We provide the corresponding standard deviations and ROC-AUCs results in Appendix A.5. The results reveal several key insights: (1) Traditional attacks incur significantly high runtime. The slight differences mainly depend on the different strategies in their attack models. (2) PIR-D achieves better accuracy among the baselines, possibly due to their consideration of permutation equivalence. AIA shows lower performance, which may be because of their limited ability to conduct attribute inference, thus affecting the classification of properties. (3) The proposed attack model outperforms all baseline methods across all datasets, achieving an average increase of 2.7% in accuracy and being 6.5× faster compared to the best baseline, demonstrating its remarkable efficiency and efficacy. The significant margin by which our method outperforms the baselines is primarily due to our specific mechanisms that ensure diversity in both reference and augmented graphs, which are essential for training a robust attack model. In contrast, conventional attacks lack such designs for shadow graph diversity, resulting in sub-optimal performance.

## 4.3 Evaluation of influencing factors (RQ2)

**Ablation study.** To ensure effectiveness, our method includes two main mechanisms: sampling diverse reference graphs and selecting diverse augmented graphs. Here, we conduct ablation studies to demonstrate their necessity, including four variants: (1) w/o structure: We discard structure-aware sampling and use simple random walks to sample reference graphs. (2) w/o selector: We discard the augmentation selector and use random removal to obtain augmented graphs. (3) w/o error: In the augmentation selector (*cf.* Eq. (6)), we ignore the approximation error and only select augmentations that maximize diversity. (4) w/o diversity: We ignore diversity in the augmentation selector (*cf.* Eq. (6)) and only select augmentations that minimize the approximation error. Figure 2 (a) shows the attack results on Facebook's node property. Notably, the complete model consistently surpasses the performance of all variants, showing the effectiveness and necessity of simultaneously sampling diverse reference graphs and selecting diverse augmented graphs.

**Hyper-parameter analysis.** We next evaluate the impact of two important hyper-parameters on our method: (1) the number of reference graphs and (2) the number of selected augmented graphs. Both directly affect the diversity of approximated models. We tune the number of reference graphs among {25, 50, 75, 100} and the number of selected augmented graphs among {2, 4, 6, 8, 10}. The results in Figure 2 (b) and 2 (c) show that as both hyper-parameters increase, the attack performance initially improves and then stabilizes. This indicates that a relatively small number of reference graphs and augmented graphs are sufficient to ensure diversity, thereby maintaining good attack performance.

## 4.4 Evaluation in different scenarios (RQ3)

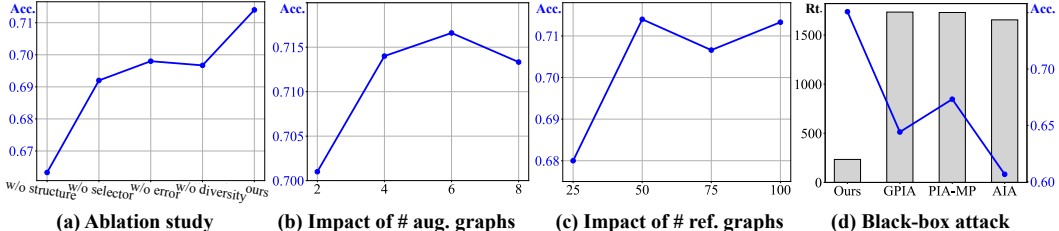

| (a) Ablation study | (b) Impact of # aug. graphs | (c) Impact of # ref. graphs | (d) Black-box attack |

Figure 2: (a) Evaluation of the necessity of considering diversity while minimizing the approximation error. (b) and (c) Impact of the number of augmented graphs (per reference graph) and reference graphs on attack accuracy, respectively. (d) Accuracy and runtime comparison in black-box settings.

To test the applicability of our method, we evaluate its performance under various conditions, including scenarios with black-box adversary knowledge, on different types of GNN models, on large-scale graph datasets, and when the target and auxiliary graphs are distinct.

**Performance on black-box knowledge.** In the black-box setting, we use model outputs, specifically posterior probabilities, to train attack models for our method and baselines. Since PIR-D and PIR-S only support white-box settings, we included another state-of-the-art black-box attack, PIA-MP [20], as detailed in Appendix A.4. The results on Facebook's node property in Figure 2 (d) show that our method improves accuracy by 11.5% compared to the best baselines while being 7.3× faster.

**Performance on other GNNs.** We conduct property inference attacks on other three fundamental GNNs: GCN [43], GAT [44], and SGC [45]. For GCN and GAT, hyper-parameters are configured according to [21], while for SGC, we set the number of hops to 2. We report the attack accuracy and runtime of our method alongside other baselines on Facebook's node property, as illustrated in Figure 3 (a)-(c). It is observed that the overall attack accuracy for SGC is comparatively lower, potentially due to the SGC model's inherent limitations in capturing property information effectively. Moreover, our method consistently achieves the highest accuracy, also demonstrating a runtime that is

Table 3: Attack comparison using distinct graphs for the target and auxiliary graphs. The arrows indicate the auxiliary graph on the left and the target graph on the right. The best results are bolded.

| | Facebook ⇒ Pokec | | | | Pokec ⇒ Facebook | | | |
| | Node | | Link | | Node | | Link | |
| | Acc. | Rt. | Acc. | Rt. | Acc. | Rt. | Acc. | Rt. |
|---|---|---|---|---|---|---|---|---|
| GPIA | 56.7 | 1732 | 54.6 | 1569 | 60.2 | 1244 | 57.5 | 1296 |
| PIR-D | 54.5 | 1794 | 56.8 | 1648 | 62.4 | 1297 | 55.9 | 1385 |
| PIR-S | 57.6 | 1777 | 52.1 | 1609 | 60.3 | 1262 | 56.1 | 1342 |
| AIA | 53.4 | 1729 | 53.0 | 1535 | 64.6 | 1237 | 55.6 | 1321 |
| Ours | **58.3** | **267** | **57.3** | **236** | **65.7** | **233** | **59.3** | **177** |

4.4× faster on GCN, 4.0× faster on GAT, and 4.3× faster on SGC compared to the best baseline.

**Performance on scalability.** We further conducted property inference attacks on a large-scale graph dataset, Pokec-100M, which contains 1,027,956 nodes and 27,718,416 edges. This graph is sampled from the original dataset [39] by retaining nodes with relatively complete features. We targeted the same node property as in the Pokec dataset, with the number of nodes in the reference graphs, shadow graphs, and target graphs set to 52,600, 50,000, and 50,000, respectively. All other settings remain consistent with previous experiments. We compare the attack accuracy and runtime of our method against other baselines. As shown in Figure 3 (d), conventional attacks incur significant computational costs on this dataset, whereas our method is 10.0× faster. Additionally, our attack accuracy is significantly higher than those of the baselines.

**Performance with distinct target and auxiliary graphs.** In the above experiments, the target and auxiliary graphs are splits of the same original graph. However, in real-world scenarios, this assumption may not hold. Therefore, we evaluate the performance of our attack under a more practical condition, where distinct graphs (from the same domain) are used as the target and auxiliary graphs. Specifically, we select Facebook and Pokec, as they are both social networks, and consider two cases: using Facebook as the target and Pokec as the auxiliary graph, and vice versa. Since the feature dimensions of these two datasets differ, the parameters of the approximated model and the target model are not directly compatible, so we apply PCA dimension reduction to align the parameters.

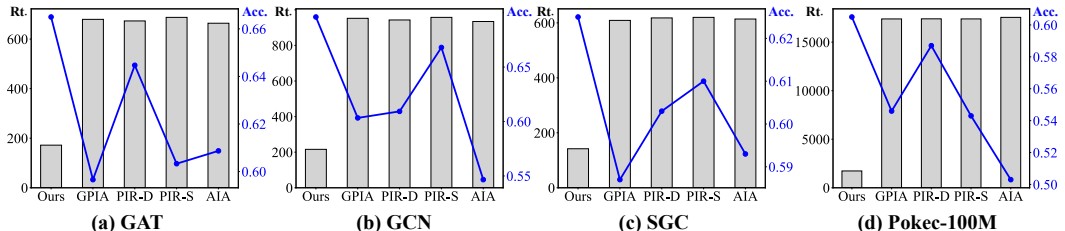

Figure 3: Comparison of average attack accuracy and runtime (seconds) on: (a)-(c) other GNNs, including GAT, GCN, and SGC; (d) a large-scale dataset, Pokec-100M.

Table 3 reports the attack accuracy and runtime. We observe that (1) the overall attack performance decreases, potentially due to the loss of property information embedded in model parameters during parameter alignment, and (2) our model consistently achieves the best performance with significant speed-ups, demonstrating its effectiveness in a more practical scenario.

## 5  Literature Review

**Property inference attack.**   The concept of property inference attack is first introduced by [23], demonstrating the leakage of sensitive properties from hidden Markov models and support vector machines in systems like speech-to-text. Building on this, attacks on various machine learning models are studied, including feed-forward neural networks, convolutional neural networks, and generative adversarial networks [22, 25, 32, 46]. Some works also consider multi-party collaborative learning scenarios [20, 24] or incorporate data poisoning [32, 47]. Specifically, [47] proposes an efficient attack based on distinguishing tests, achieving faster performance than traditional shadow training. Their setting differs from ours by the additional adversarial capability of data poisoning. Recently, with the increasing use of graphs and GNNs, security and privacy concerns are emerging [48–51]. While efforts have been made to investigate property inference attacks on GNNs [20, 21, 27], they follow the shadow training framework, which requires training a relatively large number of shadow GNN models, leading to high computational costs and reduced feasibility [47]. [52] assumes access to the embedding of whole graphs and targets at graph-level properties, which is beyond our scope.

**GNN model approximation.**   GNN model approximations are primarily based on the influence function [29, 31, 53, 54] or Newton update [30, 33]. Except for [54], these methods are utilized in the context of graph unlearning. Studies [30, 33, 54] explore model approximation for edge or node removal and analyze the corresponding approximation error bounds, yet they are limited to specific model architectures (*e.g.*, simple graph convolution, graph scattering transform). Further efforts [29, 31] extend model approximation to generic GNNs. [29] introduces a framework for edge unlearning, while [31] proposes a general unlearning framework for removing either nodes, edges, or features individually. Our model approximation differs from above by enabling the simultaneous removal of nodes and edges across generic GNN architecture. A concurrent work [53] addresses a similar model approximation as our attack; however, the additional theoretical assumptions could fail when removing a combination of nodes and edges, and their corresponding solution may significantly compromise the efficiency. Other studies [55, 56] also employ the graph shard approach. However, they may have poor efficiency in batch removal, which involves multiple retraining of sub-models.

## 6  Conclusion

In this paper, we focus on the problem of graph property inference attacks. We utilize model approximation to efficiently generate approximated models after initially training a small set of models, which replaces the costly shadow training in traditional attacks. To overcome the challenge of ensuring the diversity of approximated models while reducing the approximation error, we first derive a theoretical criterion to quantify the impact of different augmentations on approximation error. Next, we propose a diversity enhancement strategy, including a structure-aware random walk for sampling diverse reference graphs and a selection mechanism to retain optimal approximated models, utilizing edit distance to measure diversity and the theoretical criterion to assess approximation error. The retained approximated models are finally used to train an attack classifier. Extensive experiments across six real-world scenarios demonstrate our attack's outstanding efficiency and effectiveness.

## Acknowledgments and Disclosure of Funding

This work is supported by the Zhejiang Province "JianBingLingYan+X" Research and Development Plan (2024C01114).

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

# A  Appendix

## A.1  Notations

The main notations can be found in the following table.

Table 4: Description of major notations, ordered by appearance.

| Notation | Description |
|---|---|
| $G^{\mathrm{tar}}, G^{\mathrm{aux}}, \mathcal{P}(G^{\mathrm{tar}})$ | target graph, auxiliary graph, property of $G^{\mathrm{tar}}$ |
| $G^{\mathrm{ref}}, V, E$ | reference graph, its node set and edge set |
| $V^{\mathrm{R}}, E^{\mathrm{R}}, G^{\mathrm{aug}}$ | removed nodes and edges in $G^{\mathrm{ref}}$, augmented graph |
| $v, u, e$ | two nodes, an edge connects $v, u$ |
| $\mathcal{N}_l(v), \mathcal{N}_l(e)$ | the influence nodes of removing $v$ and $e$ |
| $V^{\mathrm{I}}$ | the total influenced nodes of removing $V^{\mathrm{R}}$ and $E^{\mathrm{R}}$ |
| $\theta^{\mathrm{ref}}, \theta^{\mathrm{aug}}, \ell$ | parameters of reference model/approximated model, cross-entropy loss |
| $\mathcal{G}^{\mathrm{aug}}$ | a set of augmented graphs |

## A.2  Proof of theorems

### A.2.1  Proof of Theorem 3.1

Given the GNN parameter $\theta^{\mathrm{ref}}$ on $G^{\mathrm{ref}}$, the removed nodes $V^{\mathrm{R}}$, removed edges $E^{\mathrm{R}}$ and influenced nodes $V^{\mathrm{I}}$. Assume $\ell$ is twice-differentiable everywhere and convex, we have

$$\theta^{\mathrm{aug}} \approx \theta^{\mathrm{ref}} + (\nabla^2 \sum_{v \in V/V^{\mathrm{R}}} \ell(\theta^{\mathrm{ref}}; v, E/E^{\mathrm{R}}))^{-1} \nabla(\sum_{v \in V^{\mathrm{I}} \cup V^{\mathrm{R}}} \ell(\theta^{\mathrm{ref}}; v, E) - \sum_{v \in V^{\mathrm{I}}} \ell(\theta^{\mathrm{ref}}; v, E/E^{\mathrm{R}})),$$
(7)

where $\nabla$ denote the gradient, and $\nabla^2$ denote the Hessian.

**Proof.** Let $\mathcal{L}(\theta; G^{\mathrm{aug}}) = \sum_{v \in V/V^{\mathrm{R}}} \ell(\theta; v, E/E^{\mathrm{R}})$ denote the loss function of $G^{\mathrm{aug}}$ at $\theta$, by one step newton update of $\mathcal{L}$, the approximation of $\theta^{\mathrm{aug}}$ is:

$$\theta^{\mathrm{aug}} \approx \theta^{\mathrm{ref}} - \left(\nabla^2 \mathcal{L}(\theta^{\mathrm{ref}}; G^{\mathrm{aug}})\right)^{-1} \nabla \mathcal{L}(\theta^{\mathrm{ref}}; G^{\mathrm{aug}}).$$
(8)

Let $V^{\mathrm{UI}}$ denote the uninfluenced node set, i.e., $V^{\mathrm{I}} \cup V^{\mathrm{UI}} = V/V^{\mathrm{R}}, V^{\mathrm{I}} \cap V^{\mathrm{UI}} = \emptyset$, we have

$$
\begin{aligned}
\mathcal{L}(\theta^{\mathrm{ref}}; G^{\mathrm{aug}}) &= \sum_{v \in V^{\mathrm{UI}}} \ell(\theta^{\mathrm{ref}}; v, E/E^{\mathrm{R}}) + \sum_{v \in V^{\mathrm{I}}} \ell(\theta^{\mathrm{ref}}; v, E/E^{\mathrm{R}}) \\
&= \sum_{v \in V^{\mathrm{UI}}} \ell(\theta^{\mathrm{ref}}; v, E) + \sum_{v \in V^{\mathrm{R}}} \ell(\theta^{\mathrm{ref}}; v, E) + \sum_{v \in V^{\mathrm{I}}} \ell(\theta^{\mathrm{ref}}; v, E) \\
&\quad + \sum_{v \in V^{\mathrm{I}}} \ell(\theta^{\mathrm{ref}}; v, E/E^{\mathrm{R}}) - \sum_{v \in V^{\mathrm{R}}} \ell(\theta^{\mathrm{ref}}; v, E) - \sum_{v \in V^{\mathrm{I}}} \ell(\theta^{\mathrm{ref}}; v, E) \\
&= \sum_{v \in V} \ell(\theta^{\mathrm{ref}}; v, E) + \sum_{v \in V^{\mathrm{I}}} \ell(\theta^{\mathrm{ref}}; v, E/E^{\mathrm{R}}) - \sum_{v \in V^{\mathrm{R}}} \ell(\theta^{\mathrm{ref}}; v, E) - \sum_{v \in V^{\mathrm{I}}} \ell(\theta^{\mathrm{ref}}; v, E)
\end{aligned}
$$
(9)

Given $\nabla \sum_{v \in V} \ell(\theta^{\mathrm{ref}}; v, E) = 0$, we have

$$\nabla \mathcal{L}(\theta^{\mathrm{ref}}; G^{\mathrm{aug}}) = \nabla \left( \sum_{v \in V^{\mathrm{I}}} \ell(\theta^{\mathrm{ref}}; v, E/E^{\mathrm{R}}) - \sum_{v \in V^{\mathrm{I}} \cup V^{\mathrm{R}}} \ell(\theta^{\mathrm{ref}}; v, E) \right).$$
(10)

And

$$\theta^{\mathrm{aug}} \approx \theta^{\mathrm{ref}} + \left(\nabla^2 \mathcal{L}(\theta^{\mathrm{ref}}; G^{\mathrm{aug}})\right)^{-1} \nabla \left( \sum_{v \in V^{\mathrm{I}} \cup V^{\mathrm{R}}} \ell(\theta^{\mathrm{ref}}; v, E) - \sum_{v \in V^{\mathrm{I}}} \ell(\theta^{\mathrm{ref}}; v, E/E^{\mathrm{R}}) \right),$$
(11)

which completes the proof. $\qquad\square$

### A.2.2 Proof of Theorem 3.2

Assume $\ell$ is twice-differentiable everywhere and convex, $\|\nabla \ell\|_2 \leq c_1$, $\nabla^2 \sum_{v \in V/V^{\mathrm{R}}} \ell(\theta; v, E/E^{\mathrm{R}})$ is $\gamma_1$-Lipschitz, the error bound for the approximation error is given by:

$$\|\nabla \sum_{v \in V/V^{\mathrm{R}}} \ell(\theta^{\mathrm{aug}}; v, E/E^{\mathrm{R}})\|_2 \leq C(|V^{\mathrm{R}}| + 2|V^{\mathrm{I}}|)^2 = C \cdot \delta(V^{\mathrm{R}}, E^{\mathrm{R}}), \tag{12}$$

where $|\cdot|$ denotes the cardinality of a set, and $\delta(\cdot, \cdot)$ denotes the number of nodes removed and influenced, given $V^{\mathrm{R}}$ and $E^{\mathrm{R}}$. $C$ is a constant depending on the GNN model.

**Proof.** Firstly, we consider an empirical loss for the reference model, which consists of a cross-entropy and a $L_2$-regularization:

$$\ell(\theta; v, E) = \mathrm{CE}(\theta; v, E) + \frac{\lambda}{2}\|\theta\|_2^2, \tag{13}$$

where $\mathrm{CE}(\cdot, \cdot)$ represents the cross-entropy, and we omit the ground truth for simplicity. $\lambda > 0$ denotes the $L_2$-regularization.

Let $G(\theta) = \nabla \sum_{v \in V/V^{\mathrm{R}}} \ell(\theta; v, E/E^{\mathrm{R}})$, $H_0$ denote the Hessian of $\sum_{v \in V/V^{\mathrm{R}}} \ell(\theta^{\mathrm{ref}}; v, E/E^{\mathrm{R}})$, and let

$$\Delta = \nabla \left( \sum_{v \in V^{\mathrm{I}} \cup V^{\mathrm{R}}} \ell(\theta^{\mathrm{ref}}; v, E) - \sum_{v \in V^{\mathrm{I}}} \ell(\theta^{\mathrm{ref}}; v, E/E^{\mathrm{R}}) \right). \tag{14}$$

By Taylor's Theorem, we have

$$\begin{aligned} G(\theta^{\mathrm{aug}}) &\approx G(\theta^{\mathrm{ref}} + H_0^{-1}\Delta) \\ &= G(\theta^{\mathrm{ref}}) + \nabla G(\theta^{\mathrm{ref}} + \eta H_0^{-1}\Delta)H_0^{-1}\Delta \\ &= G(\theta^{\mathrm{ref}}) + H_\eta H_0^{-1}\Delta, \end{aligned} \tag{15}$$

where $H_\eta$ denotes the hessian at $\theta_\eta = \theta^{\mathrm{ref}} + \eta H_0^{-1}\Delta$, $\eta \in [0, 1]$.

Let $G(\theta^{\mathrm{aug}}) = G(\theta^{\mathrm{ref}}) + \Delta + H_\eta H_0^{-1}\Delta - \Delta$, we have

$$\begin{aligned} G(\theta^{\mathrm{ref}}) + \Delta &= \nabla \sum_{v \in V/V^{\mathrm{R}}} \ell(\theta^{\mathrm{ref}}; v, E/E^{\mathrm{R}}) + \Delta \\ &= \nabla \sum_{v \in V^{\mathrm{UI}}} \ell(\theta^{\mathrm{ref}}; v, E/E^{\mathrm{R}}) + \nabla \sum_{v \in V^{\mathrm{I}}} \ell(\theta^{\mathrm{ref}}; v, E/E^{\mathrm{R}}) + \Delta \\ &= \nabla \sum_{v \in V} \ell(\theta^{\mathrm{ref}}; v, E) \\ &= 0. \end{aligned} \tag{16}$$

Since $G(\theta^{\mathrm{aug}}) = H_\eta H_0^{-1}\Delta - \Delta = (H_\eta - H_0)H_0^{-1}\Delta$, we have

$$\|G(\theta^{\mathrm{aug}})\|_2 = \|(H_\eta - H_0)H_0^{-1}\Delta\|_2 \leq \|H_\eta - H_0\|_2 \|H_0^{-1}\Delta\|_2. \tag{17}$$

Assume the Hessian of $\sum_{v \in V/V^{\mathrm{R}}} \ell(\theta; v, E/E^{\mathrm{R}})$ is $\gamma_1$-Lipschitz, we have

$$\begin{aligned} \|H_\eta - H_0\|_2 &= \|\nabla^2 \sum_{v \in V/V^{\mathrm{R}}} \ell(\theta_\eta; v, E/E^{\mathrm{R}}) - \nabla^2 \sum_{v \in V/V^{\mathrm{R}}} \ell(\theta^{\mathrm{ref}}; v, E/E^{\mathrm{R}})\|_2 \\ &\leq \gamma_1 \|\theta_\eta - \theta^{\mathrm{ref}}\|_2 \\ &= \gamma_1 \|\eta H_0^{-1}\Delta\|_2 \\ &\leq \gamma_1 \|H_0^{-1}\Delta\|_2, \quad \text{since } \eta \in [0, 1]. \end{aligned} \tag{18}$$

Then we have $\|G(\theta^{\mathrm{aug}})\|_2 \leq \gamma_1 \|H_0^{-1}\Delta\|_2^2$. Since

$$
\begin{aligned}
\|\Delta\|_2 &= \|\nabla \sum_{v \in V^{\mathrm{I}} \cup V^{\mathrm{R}}} \ell(\theta^{\mathrm{ref}}; v, E) - \nabla \sum_{v \in V^{\mathrm{I}}} \ell(\theta^{\mathrm{ref}}; v, E/E^{\mathrm{R}})\|_2 \\
&\leq \|\nabla \sum_{v \in V^{\mathrm{I}} \cup V^{\mathrm{R}}} \ell(\theta^{\mathrm{ref}}; v, E)\|_2 + \|\nabla \sum_{v \in V^{\mathrm{I}}} \ell(\theta^{\mathrm{ref}}; v, E/E^{\mathrm{R}})\|_2 \\
&\leq \sum_{v \in V^{\mathrm{I}} \cup V^{\mathrm{R}}} \|\nabla \ell(\theta^{\mathrm{ref}}; v, E)\|_2 + \sum_{v \in V^{\mathrm{I}}} \|\nabla \ell(\theta^{\mathrm{ref}}; v, E/E^{\mathrm{R}})\|_2 \\
&\leq (|V^{\mathrm{R}}| + 2|V^{\mathrm{I}}|)c_1, \quad \text{assume } \|\nabla \ell\|_2 \leq c_1.
\end{aligned}
\tag{19}
$$

Since $\sum_{v \in V/V^{\mathrm{R}}} \ell(\theta; v, E/E^{\mathrm{R}})$ is $\lambda(|V| - |V^{\mathrm{R}}|)$-strongly convex, we have $\|H_0^{-1}\|_2 \leq \frac{1}{\lambda(|V|-|V^{\mathrm{R}}|)}$. In practice, we keep $|V^{\mathrm{R}}|$ as a fixed value, thus $\|H_0^{-1}\|_2 \leq \frac{1}{c_2\lambda}$, where $c_2 = |V| - |V^{\mathrm{R}}|$. Finally,

$$
\begin{aligned}
\|G(\theta^{\mathrm{aug}})\|_2 &\leq \gamma_1 \|H_0^{-1}\Delta\|_2^2 \\
&\leq \gamma_1 \|H_0^{-1}\|_2^2 \|\Delta\|_2^2 \\
&\leq \frac{\gamma_1 c_1^2}{c_2^2 \lambda^2}(|V^{\mathrm{R}}| + 2|V^{\mathrm{I}}|)^2 \\
&= C(|V^{\mathrm{R}}| + 2|V^{\mathrm{I}}|)^2, \quad \text{where } C = \frac{\gamma_1 c_1^2}{c_2^2 \lambda^2}.
\end{aligned}
\tag{20}
$$

$\square$

### A.3 Training algorithm and complexity analysis

**Complexity of generating approximated models.** As the computation of gradients can be efficiently handled by the PyTorch Autograd Engine, the primary operation is solving the inverse of the Hessian (*cf.* Eq. (4)). To mitigate the high computational cost, we follow [29] in converting the inverse computation into finding the minimizer of a quadratic function, resulting in an approximated solution. By leveraging efficient Hessian-vector products and the conjugate gradient method, this can be solved with time complexity of $O(t|\theta|)$, where $|\theta|$ denotes the number of parameters, and $t$ represents the number of iterations in conjugate gradient method.

**Training algorithm.** The training algorithm for our attack is summarized in Algorithm 1.

### A.4 More experiment settings

**Details of datasets.** The statistics of the datasets used in this work are summarized in Table 5.

Table 5: Dataset statistics.

| Dataset | # nodes | # edges | # features | # classes |
|---------|---------|---------|------------|-----------|
| Pokec | 40,478 | 531,736 | 197 | 2 |
| Facebook | 4,309 | 176,468 | 1,282 | 2 |
| Pubmed | 19,717 | 88,648 | 500 | 3 |

- **Facebook** [37]: This dataset consists of 4,039 nodes and 176,468 edges. Nodes have features like birthday, education, work, name, location, gender, hometown, and language, all anonymized for privacy. The target GNN's task is to classify users' education types.
- **PubMed** [38]: This dataset includes 19,717 scientific publications related to diabetes, with a citation network of 88,648 links. Each publication is described by a TF/IDF weighted word vector from a dictionary of 500 unique words, such as male, female, children, cholesterol, and insulin. The target GNN's task is to classify the topic categories of the publications.

---
**Algorithm 1** Overall algorithm for the proposed attack.

---
**Input:** Auxilary graph $G^{\text{aug}}$, target model's parameters or posterior probabilities. Number of reference graphs $r$, number of perturbations $k$, number of selected augmented graphs $q$, weight $w$ for sampling reference graphs.
**Output:** The inferred sensitive property $\mathcal{P}(G^{\text{tar}})$ of the target graph $G^{\text{tar}}$.
1: Partition $G^{\text{aug}}$ into multiple communities using Louvain community detection.
2: Sample $r$ reference graphs using the proposed structure-aware random walk, with weight $w$.
3: **for** each reference graph $G^{\text{ref}}$ **do**
4:    Train reference model $\theta^{\text{ref}}$.
5:    $\emptyset \to \mathcal{G}^{\text{aug}}$, $\emptyset \to \text{Errors}$, $\emptyset \to \text{Perturbs}$.
6:    **for** $i$ in $1, \dots, k$ **do**
7:       Randomly select a set of nodes $V^{\text{R}}$ and a set of edges $E^{\text{R}}$ from $G^{\text{ref}}$.
8:       Add $G_i^{\text{aug}}$ into $\mathcal{G}^{\text{aug}}$.
9:       Add $(V^{\text{R}}, E^{\text{R}})$ into $\text{Perturbs}$.
10:      Compute the approximation error criterion in Eq. (5), and add it into $\text{Errors}$.
11:    **end for**
12:    Solve the optimization problem in Eq. (6), select $q$ perturbations.
13:    Calculate the $q$ approximated models by Eq. (4).
14:    Save the $q$ parameters (or posterior) and the $q$ properties of the selected augmented graphs.
15: **end for**
16: Train an attack model based on the $r \cdot q$ parameters (or posterior) and properties, classify $\mathcal{P}(G^{\text{tar}})$ of the target graph $G^{\text{tar}}$.

---

- **Pokec** [39]: This online social network dataset is from Slovakia. Each node has anonymized features such as gender, age, and hobbies. We follow [21] to sample nodes with relatively complete features, resulting in a graph with 40,478 nodes and 531,736 edges, using gender, age, height, weight, and region as node features. The target GNN's task is to classify whether a user's all friendships are public.

**Details of sensitive properties.** For each dataset, we design one node property and one link property to be targeted in our attacks. For Facebook and Pokec we select gender as the property attribute. For PubMed, we select the keyword "Insuli" as the property feature, as it has the highest TF-IDF weight. These properties are summarized in Table 1.

**Statistics of reference, shadow, and target graphs.** The sizes and numbers of reference graphs we use are summarized in Table 6.

Table 6: Numbers and sizes of reference graphs.

| Dataset | Ref. graph number | Ref. graph size |
|---------|-------------------|-----------------|
| Pokec | 50 | 525 |
| Facebook | 50 | 3200 |
| Pubmed | 100 | 4250 |

For all baselines, we follow the settings specified in [21] to sample shadow graphs: the size of each shadow graph is 20%, 25%, and 30% of Pokec, Facebook, and Pubmed, respectively, and the number of shadow graphs is 700 for all datasets.

For target graphs, we sample 300 shadow graphs for each dataset; the size of each shadow graph is 20%, 25%, and 30% of Pokec, Facebook, and Pubmed datasets, respectively. To ensure fairness, we evaluate our method and baselines on the same target graphs.

**More implementation details.** All experiments are conducted on a machine of Ubuntu 20.04 system with AMD EPYC 7763 (756GB memory) and NVIDIA RTX3090 GPU (24GB memory). All models are implemented in PyTorch version 2.0.1 with CUDA version 11.8 and Python 3.8.0.

The attack model is trained for 100 epochs with a learning rate of 1e-3 and a weight decay of 5e-4. We use cross-entropy loss as the loss function and Adam optimizer.

**Baselines for white-box setting.**

- **PIR-S** [22]: A property inference attack considering permutation equivalence in feed-forward neural networks, using node permutations on the hidden layers of a fully connected neural network.
- **PIR-D** [22]: Similar to PIR-S, permutation equivalence is achieved by ensuring all permutations of any layer have the same set-based representation.

**Baselines for black-box setting.**

- **PIA-MP** [20]: An attack method designed for multi-party machine learning. The primary difference between PIA-MP and GPIA is that in the former, all shadow models (as well as the target model) are queried on a fixed dataset to obtain output posterior probabilities.

**Baselines for both settings.**

- **GPIA** [21]: A method designed for graphs and GNNs in both white-box and black-box settings, following the traditional attack framework. The inferred properties include node properties and link properties.
- **AIA** [42]: We conduct an attribute inference attack that predicts the property attributes by accessing the parameters or posteriors. We then evaluate the property inference performance based on the predicted values of the property attributes.

### A.5 More experiment results

Table 7: Standard deviation.

| Standard deviation | Facebook | | Pubmed | | Pokec | |
|---|---|---|---|---|---|---|
| | Node | Link | Node | Link | Node | Link |
| GPIA | 0.9 | 2.1 | 0.7 | 0.5 | 0.5 | 0.3 |
| PIR-D | 2.9 | 1.5 | 3.2 | 1.7 | 1.4 | 2.3 |
| PIR-S | 1.6 | 2.7 | 1.7 | 1.7 | 0.9 | 2.1 |
| AIA | 2.3 | 2.6 | 5.3 | 0.2 | 0.5 | 1.2 |
| Ours | 0.5 | 0.2 | 2.0 | 0.4 | 0.7 | 1.0 |

**Standard deviations for Table 2.** Table 7 reports the standard deviations corresponding to the average accuracies in Table 2.

**ROC-AUC results.** We also report the ROC-AUCs of the proposed attack method compared to other baseline methods on the six sensitive properties, as shown in Table 8. The results demonstrate that our model can consistently achieve the best ROC-AUC result, confirming its notable effectiveness.

Table 8: ROC-AUC comparison of our method and baselines. "Node" and "Link" denote node and link properties respectively. The best results are highlighted in bold.

| ROC-AUC | Facebook | | Pubmed | | Pokec | |
|---|---|---|---|---|---|---|
| | Node | Link | Node | Link | Node | Link |
| GPIA | 50.4±0.7 | 46.4±1.4 | 69.8±5.5 | 57.4±0.7 | 56.8±0.4 | 54.1±0.4 |
| PIR-D | 62.8±1.3 | 48.6±0.9 | 83.0±2.5 | 62.0±5.1 | 60.5±1.8 | 60.9±1.4 |
| PIR-S | 35.9±3.6 | 49.4±1.4 | 73.7±0.6 | 61.9±2.3 | 60.7±1.2 | 59.9±1.6 |
| AIA | 48.5±2.3 | 52.8±1.1 | 61.4±4.0 | 48.2±3.7 | 53.0±3.4 | 51.3±1.8 |
| Ours | **64.0±0.3** | **53.4±0.4** | **87.1±2.7** | **69.3±8.7** | **63.0±1.1** | **61.6±0.5** |

### A.6 Limitation and future work

In this work, we consider two settings: white-box and black-box, which encompass many real-world scenarios. However, stricter cases exist where the attacker can make only a limited number of queries or access only model predictions (*i.e.*, classification results). We acknowledge that our method does not yet address these cases. Additionally, some studies explore scenarios where attackers have enhanced capabilities, such as data poisoning. We leave the investigation of efficient attacks under these conditions for future work.

### A.7 Potential impacts

While the proposed method is designed to infer properties of specific graph data, our primary objective is to raise awareness of the privacy and security concerns associated with GNNs and to encourage the implementation of protective measures in model design. Traditional property inference methods are often inefficient, and despite efforts to illuminate potential threats, less practical attack scenarios may not receive adequate attention. Nonetheless, the privacy risks persist. We seek to bring this threat to the forefront and advocate for the adoption of more robust protective measures.

