# OpenReview forum: "Can Graph Neural Networks Expose Training Data Properties? An Efficient Risk Assessment Approach"
_NeurIPS.cc/2024/Conference — NeurIPS 2024 poster_

### Official Review · Reviewer_5YZA · 2024-06-26

**Soundness:** 3
**Presentation:** 3
**Contribution:** 3
**Rating:** 6
**Confidence:** 4

**Summary:**

The paper introduces an improved method for conducting property inference attacks on graph neural networks (GNNs), particularly focusing on reducing the computational overhead associated with traditional approaches that rely on numerous shadow models. The authors propose a model approximation technique to generate a sufficient number of approximated models for attacks, without the need for extensive retraining. They introduce a diversity-enhancing mechanism to ensure the effectiveness of the approximations. Experiments demonstrate notable improvements in attack accuracy and efficiency compared to existing methods, validating the proposed approach's effectiveness and efficiency.

**Strengths:**

- Steady theretical support
- Well-written paper
- Novel combination of different algorithms

**Weaknesses:**

- Lack of details and certain explanations
- Weak experimental settings

**Questions:**

- This paper is well-structured and well-written. I particularly appreciate the authors' efforts in providing thorough theoretical explanations and introducing innovative approaches. Replacing the training of numerous shadow models with model approximation is a significant and impactful contribution, effectively enhancing the efficiency of general property inference attacks.

- This paper assumes that the target and auxiliary graphs are splits of the same original graph, which is an impractical assumption in real-world scenarios. It would strengthen the paper's applicability if the authors used distinct network graphs (yet from the same dimain) for the target and auxiliary graphs.

- When comparing training efficiency with state-of-the-art methods, it is unclear whether the time required to generate diverse approximated models and calculate error is included. Including such details would enhance the transparency and reliability of the comparisons.

- While the paper presents improvements over baselines, it does not thoroughly discuss why the proposed method might outperform baselines by a big margin. Although it is clear that using a few approximated models in place of hundreds of shadow models contributes to efficiency, the reasons why it is much more effective than the baselines need to be further discussed.

- The choice of property attributes inferred in this study (e.g., # male users > # female users, or # publications with "IS" > # publications without "IS") seems kind of limited. The significance of this work would be enhanced if the authors expanded their experiments to include the inference of a broader variety of node or link properties such as the portion of different kinds of nodes or links.

- I would be pleased to increase my score if the authors address these issues, which would substantially enhance the paper's contribution to the field.

**Limitations:**

see questions

---

> ### Author Rebuttal · Authors · 2024-08-07
>
> >Q2: Experiment using distinct network graphs for the target and auxiliary graphs.
>
> Thank you for your insightful comment. We select two social network datasets, Facebook and Pokec. We consider two cases: Facebook and Pokec as the target and auxiliary graphs respectively, and vice versa. We maintain other settings unchanged and report the attack accuracy and runtime below. While the overall attack performances decline, our model consistently performs best with significant speed-ups.
>
> Facebook $\Rightarrow$ Pokec (Node property)
> |         | GPIA | PEPIA-DS | PEPIA-S | AIA | Ours     |
> |---------|------|----------|---------|-----|----------|
> | **Accuracy** | 56.7 | 54.5     | 57.6    | 53.4| **58.3** |
> | **Runtime(s)**   | 1732 | 1794     | 1777    | 1729| **267**  |
>
> Facebook $\Rightarrow$ Pokec (Link property)
> |         | GPIA | PEPIA-DS | PEPIA-S | AIA | Ours     |
> |---------|------|----------|---------|-----|----------|
> | **Accuracy** | 54.6 | 56.8     | 52.1    | 53.0| **57.3** |
> | **Runtime(s)**   | 1569 | 1648     | 1609    | 1535| **236**  |
>
> Pokec $\Rightarrow$ Facebook (Node property)
> |         | GPIA | PEPIA-DS | PEPIA-S | AIA | Ours     |
> |---------|------|----------|---------|-----|----------|
> | **Accuracy** | 60.2 | 62.4     | 60.3    | 64.6| **65.7** |
> | **Runtime(s)**   | 1244 | 1297     | 1262    | 1237| **233**  |
>
> Pokec $\Rightarrow$ Facebook (Link property)
> |         | GPIA | PEPIA-DS | PEPIA-S | AIA | Ours     |
> |---------|------|----------|---------|-----|----------|
> | **Accuracy** | 57.5 | 55.9     | 56.1    | 55.6| **59.3** |
> | **Runtime(s)**   | 1296 | 1385     | 1342    | 1321| **177**  |
>
>
> >Q3: Details of runtime.
>
> Thank you for this valuable suggestion. We will update our manuscript accordingly. The reported runtime covers the entire attack process, including generating approximated models and calculating errors. As an example, We provide a runtime analysis of the attack on Facebook’s node property:
> |Task|Time(s)|
> |-|-|
> | Sampling reference graphs| 12       |
> | Training reference models  | 74       |
> | Generating augmented graphs, computing errors and diversity, and selecting augmentations | 55       |
> | Generating approximated models    | 103      |
> | Inferring target graphs' properties    | 10       |
> | Total    | 254       |
>
>
> >Q4: Discussion on the performance.
>
> Thanks for your reminder. The diversity of graphs used to generate parameters or posteriors is crucial for training a strong attack model [1,2]. Our approach involves specific mechanisms to ensure diversity in both reference and augmented graphs, greatly boosting attack performance as shown in our ablation study. In contrast, conventional attacks do not incorporate specific designs for the diversity of shadow graphs, thus leading to sub-optimal performance. We will incorporate this discussion into our paper.
>
> >Q5: Experiments on other properties.
>
> As suggested, we evaluated our approach on the OGBN products dataset [3]. Products are categorized into consumer goods and non-consumer goods based on label descriptions. We define the node property as the proportion of non-consumer goods in the target graphs: 35% (original) or 65% (higher); other settings maintain consistency with our manuscript. The table below presents our results compared to the best baseline. Our attack achieves superior performance with speeds of  12.0× faster, showing its effectiveness across a broader range of properties.
>
> | **Method** | **Accuracy** | **ROC-AUC** | **Runtime(s)** |
> |------------|--------------|-------------|----------------|
> | PEPIA-DS   | 92.3         | 88.5        | 34881          |
> | Ours       | **93.2**     | **89.1**    | **2918**       |
>
>
> [1] Property Inference Attacks on Fully Connected Neural Networks using Permutation Invariant Representations
>
> [2] Formalizing and Estimating Distribution Inference Risks
>
> [3] https://ogb.stanford.edu/docs/nodeprop/#ogbn-products

---

> ### Author Response · Authors · 2024-08-11
>
> Dear Reviewer,
>
> We appreciate your recognition of the importance of our work and the time you took to provide a detailed review. We would like to confirm that we have addressed all your concerns about our submission. If there is anything more we can do to help you in evaluating our paper, please don't hesitate to let us know.
>
> Best regards, Authors

---

> ### Author Response · Authors · 2024-08-14
>
> As we are nearing the end of the discussion phase, we would like to kindly inquire if you have any remaining questions about our paper. We would appreciate any feedback and would be glad to discuss it further.

---

### Official Review · Reviewer_xZDY · 2024-07-10

**Soundness:** 2
**Presentation:** 4
**Contribution:** 2
**Rating:** 5
**Confidence:** 4

**Summary:**

This paper delivers an effective way to improve the efficiency of property inference attacks for GNNs. Instead of training numerous shadow models, the authors propose to train a few reference models and use an efficient approximation method to obtain other shadow models trained on slightly augmented shadow graphs. Experimental results show that the proposed property inference attack achieves better performance in a shorter time.

**Strengths:**

- To me, this paper finds an interesting application of machine unlearning in reducing the time complexity of training shadow models in inference attacks.
- The experimental results show a desirable performance of the proposed method over baseline attack methods in both efficacy and efficiency.
- The paper is overall well-organized and easy to follow.

**Weaknesses:**

- The scope and significance of this paper are somewhat limited. The main contribution of this paper is to improve the efficiency of a specific inference attack model. It would be helpful to provide more discussions on the new attack algorithm, e.g., insights on defending against the new attack method.
- The technical novelty is limited. In my opinion, the technical part of this paper is pretty much a certified graph unlearning method, which has been studied in previous literature [1,2,3,4]. I don't find a distinct improvement or new contribution of the technical part compared with [1,2,3,4].
- The inverse Hessian in influence computation is seen as highly consuming. It would be helpful to add some complexity analysis of the algorithm.

[1] Wu, Kun, et al. "Certified edge unlearning for graph neural networks." KDD 2023.

[2] Pan, Chao, Eli Chien, and Olgica Milenkovic. "Unlearning graph classifiers with limited data resources." TheWebConf 2023.

[3] Chien, Eli, Chao Pan, and Olgica Milenkovic. "Efficient model updates for approximate unlearning of graph-structured data." ICLR 2023.

[4] Wu, Jiancan, et al. "Gif: A general graph unlearning strategy via influence function." TheWebConf 2023.

**Questions:**

The theoretical analysis still relies on the convexity assumptions. To satisfy the assumptions, it would be feasible to use a linearized reference model. It is not clear which type of GNN is used as the reference GNN.

**Limitations:**

The authors have included detailed discussions on the societal impact of their work.

---

> ### Author Rebuttal · Authors · 2024-08-07
>
> > W1: Limited scope and significance.
>
> Thanks for your comment.
>
> We'd like to emphasize that inefficiency is a major bottleneck in graph property inference attacks. Our contribution significantly enhances the efficiency of these attacks, enabling their practical application at scale, which is essential for both academic research and real-world use. Specifically, existing attacks can take over 3 days on graphs with more than 1,000,000 nodes using 700 shadow models [1], making them practically infeasible. Our method can reduce this to a few hours. Even on medium-large graphs with 200,000 nodes, current attacks take up to 7.5 hours, while our attack is over 10x faster, requiring less than one-tenth of the shadow models trained from scratch. Additionally, our attack also delivers superior performance.
>
> We also acknowledge the importance of discussing defensive methods against these more efficient attacks. Our attack can facilitate the faster development of defense strategies. Effective defense hinges on its ability to counteract attacks, and a more effective and efficient attack can serve as a tool to refine defense strategies.  For example, adversarial learning [2] iteratively adjusts both defense and attack strategies to minimize the success rate of attacks. Due to the inefficiency, adopting conventional property inference in adversarial learning may lead to poor defense efficiency. In contrast, our attack has the potential to accelerate the defense process, enhancing its feasibility.
>
> >W2: Technical novelty is limited.
>
> Thank you for your comment, which inspired us to clarify our contribution and the differences between our technique and existing methods.
>
> First, our approximation technique differs from existing graph unlearning methods in two key ways:
>
> (1) Unlike existing graph unlearning methods, which are restricted to removing either nodes, edges, or features individually [3,4], or are limited to specific GNN architectures [5,6], our method enables the removal of both nodes and edges across any GNN architecture. This broad applicability significantly enhances its utility in creating diverse augmented graphs, thereby improving attack performance across a wide range of target GNNs.
>
> (2) Unlike existing approaches that rely on predefined removals, we introduce a novel selection mechanism for determining which nodes and edges to remove, ensuring diverse augmented graphs and accurate approximation. This facilitates the creation of a strong attack model. Our selection comprises three parts:
>
> - To measure the accuracy of model approximations, We derive a theoretical error estimation with innovative mathematical derivations, which is computable given the removal. As we remove both nodes and edges with generic GNNs, existing analyses [3, 5, 6] can not directly apply.
> - To measure the total diversity of graphs through various removals, we propose using the efficient edit distance as the metric.
> - To minimize approximation error while maximizing total diversity, we formulate this optimization as a quadratic integer programming problem, which is efficiently solvable.
>
> Furthermore, we emphasize that the biggest contribution of this work is improving the efficiency of graph property attacks, enabling practical application (please refer to our response in W1). To achieve this, we designed two novel technical components: (1) the diverse sampling of reference graphs, which is beyond the scope of unlearning, and (2) the selection of augmented graphs with model approximation, which differs from existing unlearning in both the type and selection of removals. These components form the core technical contributions of this work.
>
> > W3: Computation of inverse Hessian.
>
> The complexity of model approximation is relatively light. As the gradient can be easily computed by Pytorch Autograd Engine, the main operation is solving the inverse of Hessian. We follow [3] to reduce this to linear complexity by converting inverse computation into finding a unique minimizer of a quadratic function. With the efficient hessian-vector product and the conjugate gradient method (CG), this can be solved with $O(t|\theta|)$ time complexity [3], where $|\theta|$ denotes the number of parameters and $t$ denotes the iteration number in CG. In experiments, our attack performs well with few iterations. We will update our paper to include these computational details.
>
> >Q1: Convexity assumptions.
>
> Thank you for your insightful comment. To the best of our knowledge, no generic unlearning approach with theoretical guarantees has successfully removed the convexity assumption, which is a challenging task. Due to the non-convex nature of GNN models, the Hessian can become non-invertible. To address this, we adopted a common solution of introducing a damping term to the Hessian, which has proven effective in [3,7].
>
> In our experiments, reference GNNs use the same architectures as the target models, including GraphSAGE, GCN, and GAT. To better align with theoretical assumptions, we also test on SGC, a linear GNN, with 2 hops. We compare attack performance and runtime on Facebook's node property, with results detailed below.
>
> ||**GPIA**|**PEPIA-DS**|**PEPIA-S**|**AIA**|**Ours**|
> |-|:-:|:-:|:-:|:-:|:-:|
> |**Accuracy**|58.7|60.3|61.0|59.3|**62.5**|
> |**Runtime(s)**|609|618|620|614|**142**|
>
> The results show that our method consistently outperforms other baselines with exceptional efficiency on the linear GNNs.
>
> [1] Group Property Inference Attacks Against Graph Neural Networks
>
> [2] Adversarial learning techniques for security and privacy preservation: A comprehensive review
>
> [3] Certified edge unlearning for graph neural networks
>
> [4] Gif: A general graph unlearning strategy via influence function
>
> [5] Unlearning graph classifiers with limited data resources
>
> [6] Efficient model updates for approximate unlearning of graph-structured data
>
> [7] Understanding Black-box Predictions via Influence Functions

---

> ### Author Response · Authors · 2024-08-11
>
> Dear Reviewer,
>
> Thank you for your review and comments. In our rebuttal, we made special efforts to clarify the distinctions between our approach and existing graph unlearning methods, provided additional discussions on our attack algorithm, and enhanced our technical details. We would greatly appreciate any feedback on any existing or new points we may not have covered, and we would be glad to address or discuss them further.
>
> Best regards, Authors

---

> ### Comment · Reviewer_xZDY · 2024-08-12
>
> I thank the authors for their further clarification which helps strengthen the contributions of this work. However, I still think the technical novelty of the model approximation part (which also means the theoretical contribution) is somewhat limited. The mentioned simultaneous removal of nodes, edges, and features and removal for various GNN architectures seem also to be covered by existing studies [1]. Further comparison and clarification would be helpful. Based on the current response, I am willing to improve my score to 4.
>
> [1] Dong, Yushun, et al. "IDEA: A Flexible Framework of Certified Unlearning for Graph Neural Networks." KDD 2024.

---

> ### Author Response · Authors · 2024-08-12
>
> Thank you for bringing this work to our attention. We note that this work was published online in July of this year, which was subsequent to the submission of our paper. Therefore, we did not discuss it in our initial version. After a thorough review of this paper, we have identified two key aspects where our approach differs significantly:
>
> (1) The approach in [1] relies on additional theoretical assumptions that impact the efficiency of model approximation, making it less suitable for our efficiency requirements. Specifically, they define different affected node groups based on the type of unlearning requests (e.g., node removal, edge removal), with the assumption that these groups do not overlap. When removing a mix of nodes and edges, this assumption may fail. To address this, they split the removal into multiple sets, each meeting the non-overlapping requirement, and perform the unlearning processes on these sets sequentially. This approach incurs significantly higher computational costs; for instance, if the removal is divided into five sets, five separate unlearning processes are required sequentially. In contrast, our approach groups nodes simply into influenced set and removed set, requiring only one unlearning process per removal, which ensures greater efficiency.
>
> (2) We employ a different error estimation approach. Specifically, the error estimation proposed in [1] supports their certified unlearning proof, which involves the more complex computation of the inverse Hessian. This could significantly compromise the speed of selecting removals in our attack. In comparison, our error estimation can be computed directly based on the removal and graph structure, allowing for a more efficient attack.
>
> [1] Dong, Yushun, et al. "IDEA: A Flexible Framework of Certified Unlearning for Graph Neural Networks." KDD 2024.

---

> > ### Comment · Reviewer_xZDY · 2024-08-13
> >
> > I thank the authors for their further response. I agree with the first point about the shortcomings of the IDEA method, but I don't get the difference between the two methods in the inverse Hessian computation step. In addition, the computation of the inverse Hessian is not claimed to be an original contribution by this work. Nevertheless, it is true that IDEA was released recently and can be seen as a concurrent work, so it is not a big deal.
> >
> > I suggest the authors add more analyses and discussions on the efficiency of the proposed model approximation part. In comparison, Theorem 2 is not quite informative here as the advantage of the proposed model approximation compared with previous graph unlearning methods is its efficiency. If more details are added, I would like to further raise my score.

---

> ### Author Response · Authors · 2024-08-13
>
> We apologize for any confusion caused. We want to clarify that the proposed error estimation does not accelerate the model approximation itself but rather speeds up the overall attack process. Specifically, in our attack framework, we need to select augmented graphs that incur relatively lower approximation errors, enabling a more effective attack (as shown in our ablation study). The efficiency we refer to is in this selection step. Since we need to measure the error for each augmented graph, the error measurement should be straightforward and easy to compute. Therefore, we introduce a new error bound in Theorem 2, which supports the removal of both nodes and edges. This bound is then used as an efficient error estimation to select augmented graphs.
>
> Upon careful review, we found that the IDEA method also derived a similar error bound. However, it involves the term of the inverse Hessian (Eq. (16) of [1]),  which is more complex and cannot be easily computed for all augmented graphs, making it unsuitable for our selection step.
>
> In summary, our goal is not to achieve faster model approximation, but rather to use our error bound to help efficiently select data, which in turn makes the overall attack process faster.
>
> [1] Dong, Yushun, et al. "IDEA: A Flexible Framework of Certified Unlearning for Graph Neural Networks." KDD 2024.

---

> > ### Comment · Reviewer_xZDY · 2024-08-13
> >
> > I thank the authors for providing further clarifications. Most of my concerns are addressed. It would be great if the authors could add the discussions in the rebuttal phase to the revised paper.

---

> ### Author Response · Authors · 2024-08-14
>
> We are happy to hear that our response was helpful. Thank you for your prompt feedback and recognition. We will revise our paper according to your suggestions.

---

### Official Review · Reviewer_j8ts · 2024-07-14

**Soundness:** 3
**Presentation:** 2
**Contribution:** 2
**Rating:** 5
**Confidence:** 3

**Summary:**

This paper proposes a more efficient property inference attack on graph neural networks (GNNs). Particularly, it uses model approximation methods to reduce the number of shadow models that needs to be trained by the graph property inference attack. Here only a limited number of shadow models are trained from scratch while model approximation method is used to generate other approximated shadow models without training them. Theoretical upper bounds for the approximation error are also provided.

**Strengths:**

1.	The paper is overall clearly written and easy to follow.
2.	The research problem of making the inference attacks more efficient is relevant.
3.	The proposed method achieves the SoTA attack performance and largely reduces the training time of the attack model.

**Weaknesses:**

1.	How scalable is this attack for larger datasets? Current experiments are limited. Please perform the attack on larger datasets from the OGBN dataset (ArXiv, Products)
2.	How well does this scale for other GNNs? Please evaluate with other GNN models.
3. Fig. 1 is confusing. Please update it to clearly show the difference between the proposed approach and the existing attacks.

**Questions:**

Please refer to the weaknesses.

**Limitations:**

Sufficiently addressed.

---

> ### Author Rebuttal · Authors · 2024-08-07
>
> > W1: Experiments on larger datasets.
>
> As noted in Appendix A.5 and Table 8, we have already included experiments on a million-level dataset  Pokec-100M, which contains 1,027,956 nodes and 27,718,416 edges. We found that the best baseline, PEPIA-DS, incurs a significant cost, while our method is 10.2× faster and achieves both higher accuracy and ROC-AUC. To ensure that the results highlighting scalability are emphasized, we have restructured our paper to mention this experiment at the beginning of the experiments section.
>
> Additionally, as you suggested, we further evaluate our approach on the OGBN products dataset (2,449,029 nodes), larger than Pokec-100M. We categorize products into consumer goods and non-consumer goods based on label descriptions. The property is set as the portion of non-consumer goods in the target graphs: 35\% (original) or 65\% (higher); the rest settings remain consistent with our manuscript. As shown in the table below,  our attack achieves superior performance compared to the best baseline, with speeds of 12.0× faster, underscoring its scalability.
>
>
> | **Method** | **Accuracy** | **ROC-AUC** | **Runtime(s)** |
> |------------|--------------|-------------|----------------|
> | PEPIA-DS   | 92.3         | 88.5        | 34881          |
> | Ours       | **93.2**     | **89.1**    | **2918**       |
>
>
> >W2: Experiments on other GNNs.
>
> As indicated in Appendix A.5 and demonstrated in Figure 3, we have already expanded our experimental scope to include attacks on other GNNs, including GCN and GAT.  To highlight these results, we have restructured our paper to mention this experiment at the beginning of the experiments section.
>
> Additionally, to show the significance on other types of GNNs, we further evaluate with SGC [1], setting the number of hops to 2.
> The table below shows the performance and runtime of the attack on Facebook's node property, using SGC as target models.
> The results demonstrate our method consistently demonstrates superior performance and exceptional efficiency on these widely adopted GNNs.
>
> |            | **GPIA** | **PEPIA-DS** | **PEPIA-S** | **AIA** | **Ours** |
> |------------|----------|--------------|-------------|---------|----------|
> | **Accuracy** | 58.7    | 60.3         | 61.0        | 59.3    | **62.5** |
> | **Runtime(s)** | 609    | 618          | 620         | 614     | **142**  |
>
> >W3: Fig.1 is confusing.
>
> Thank you for your valuable comment.
> We have revised Fig.1 to clearly show the difference between our method and the existing attacks. Please refer to Fig. 1 in our uploaded rebuttal PDF. We will incorporate this new figure into our paper.
>
> [1] Simplifying Graph Convolutional Networks

---

> ### Author Response · Authors · 2024-08-11
>
> Dear Reviewer,
>
> Thank you for your review and comments. We hope our rebuttal and additional evaluations have effectively addressed your primary concerns regarding our paper. We would greatly appreciate any feedback on any existing or new points we may not have covered, and we would be glad to address or discuss them further.
>
> Best regards, Authors

---

> ### Comment · Reviewer_j8ts · 2024-08-13
>
> I thank the authors for their rebuttal. I am increasing my score to 5.

---

> ### Author Response · Authors · 2024-08-14
>
> We are happy to hear that our response was helpful. Thank you for your prompt feedback and recognition.

---

### Author Rebuttal · Authors · 2024-08-07

We would like to thank all the reviewers for their time and effort in reviewing our paper. As suggested by Reviewer j8ts, the following PDF contains our revised figure to better illustrate the differences between our method and existing attacks.

---

### Decision · Program_Chairs · 2024-09-25

**Decision:**

Accept (poster)

**Comment:**

This paper proposes an efficient property inference attack on graph neural networks (GNNs). The proposed method achieves the SOTA attack performance. The paper is well-written and has provided theoretical support. Most of the concerns from the reviewers have been addressed by the authors' rebuttal. Therefore, I recommend accept.